# Therapeutic Efficacy of Intranasal *N*-Acetyl-L-Cysteine with Cell-Penetrating Peptide-Modified Polymer Micelles on Neuropathic Pain in Partial Sciatic Nerve Ligation Mice

**DOI:** 10.3390/pharmaceutics17010044

**Published:** 2025-01-01

**Authors:** Hiroshi Nango, Ai Takahashi, Naoto Suzuki, Takumi Kurano, Saia Sakamoto, Taiki Nagatomo, Toyofumi Suzuki, Takanori Kanazawa, Yasuhiro Kosuge, Hiroko Miyagishi

**Affiliations:** 1Laboratory of Pharmacology, School of Pharmacy, Nihon University, 7-7-1 Narashinodai, Funabashi 274-8555, Japan; nango.hiroshi@nihon-u.ac.jp (H.N.); phai22002@g.nihon-u.ac.jp (A.T.); phsa18114@g.nihon-u.ac.jp (S.S.); 2Laboratory of Pharmaceutics, School of Pharmacy, Nihon University, 7-7-1 Narashinodai, Funabashi 274-8555, Japan; suzuki.naoto65@nihon-u.ac.jp (N.S.); phta19001@g.nihon-u.ac.jp (T.K.); nagatomo.taiki@nihon-u.ac.jp (T.N.); suzuki.toyofumi@nihon-u.ac.jp (T.S.); 3Department of Clinical Pharmacology, Graduate School of Biomedical Sciences, Tokushima University, 1-78-1 Shoumachi, Tokushima 770-8505, Japan; kanazawa@tokushima-u.ac.jp

**Keywords:** intranasal administration, cell-penetrating peptide-modified carrier, *N*-acetyl-L-cysteine, neuropathic pain, mechanical allodynia, partial sciatic nerve ligation, spinal dorsal horn, microglia

## Abstract

**Background/Objectives**: We previously demonstrated that the intranasal administration of cell-penetrating Tat peptide-modified carrier, PEG-PCL-Tat, improves drug delivery to the central nervous system. This study aimed to evaluate the potential of the post-onset intranasal administration of *N*-acetyl-L-cysteine (NAC) combined with PEG-PCL-Tat (NAC/PPT) for neuropathic pain. **Methods**: Neuropathic pain was induced by partial sciatic nerve ligation (PSNL) in mice. Mechanical allodynia was assessed using the von Frey test on days 11–14 post-ligation. NAC or NAC/PPT was intranasally administered after pain onset. Western blotting and immunohistochemistry were conducted to evaluate ionized calcium-binding adapter molecule 1 (Iba-1) expression and microglial activation in the spinal cord. **Results**: Mechanical allodynia was exacerbated 11 days after the ligation in PSNL mice. The intranasal administration of NAC alone prevented allodynia exacerbation but failed to provide a therapeutic effect against allodynia in PSNL mice. In contrast, NAC/PPT administration ameliorated PSNL-induced tactile allodynia, with maximum efficacy seen 13 and 14 days after ligation. Western blotting demonstrated that Iba-1 levels tended to increase in PSNL mice compared to controls. This trend of increased Iba-1 levels in PSNL mice was attenuated by the administration of NAC/PPT, but not by NAC alone. Immunohistochemistry revealed an increased number of Iba-1-stained microglia in the ipsilateral spinal cord of PSNL mice, which were significantly suppressed by the administration of NAC/PPT. **Conclusions**: These results suggest that the post-onset intranasal administration of NAC/PPT ameliorates mechanical allodynia by suppressing microglia induction and that intranasal delivery with PEG-PCL-Tat might be a useful tool for the pharmacological management of neuropathic pain.

## 1. Introduction

Neuropathic pain is defined a chronic pain state caused by a lesion or disease affecting the somatosensory nervous system, and its prevalence is estimated to range from 6.9 to 10% in the general population [1,2,3]. Its diverse etiologies, including stroke, spinal cord injury, diabetic neuropathy, and postherpetic neuralgia [4], hamper preventive strategies due to complexities introduced by unpredictability and various underlying conditions [5]. Therefore, therapeutic interventions are typically initiated after symptom onset, underscoring the importance of post-onset treatment strategies.

Mechanical allodynia, a hallmark symptom of neuropathic pain, impairs quality of life by transforming normally innocuous tactile stimuli into painful sensations [6]. This transformation is linked to abnormal functions within the spinal dorsal horn circuitry [7], as evidenced by increased neuronal activity in the spinal dorsal horn following mild tactile stimulation in spared nerve injury-induced neuropathic pain model mice [8]. Additionally, recent findings indicate that interactions between neurons and glial cells, including astrocytes, microglia, and oligodendrocytes, are crucial for neuropathic pain responses [9,10,11,12]. Therefore, mitigating the pathophysiological changes in neurons and glial cells within the spinal dorsal horn is a promising therapeutic strategy for neuropathic pain.

*N*-Acetyl-L-cysteine (NAC) exhibits potent antioxidant properties by serving as a direct scavenger of reactive oxygen species (ROS) and a precursor of the antioxidant glutathione [13]. In vitro studies have revealed that NAC protects neurons and glial cells from oxidative stress [14,15,16,17] and the harmful effects of tumor necrosis factor-alpha exposure [18,19,20,21]. Despite its beneficial properties, radiolabeled or stable isotope-labeled NAC is notably absent in the brain and spinal cord after intravenous injection [22,23], suggesting potential challenges in achieving therapeutic concentrations in the central nervous system. Nevertheless, NAC has shown promise in terms of alleviating neuropathic pain. Intraperitoneal NAC administration (150 mg/kg) in chronic constriction injury (CCI)-induced neuropathic pain model rats mitigated oxidative stress in the lumbosacral spinal cord [24] and the improved mechanical threshold [25]. Furthermore, oral NAC administration (100 mg/kg) in rats attenuated mechanical allodynia by suppressing CCI-induced microglia activation in the spinal dorsal horn [26]. These studies suggest that the alleviation of neuropathic pain by NAC is significantly attributed to its action on the spinal cord. However, considering the limited blood–brain barrier permeability of NAC, the development of novel drug delivery systems to improve the penetration of NAC into the brain and spinal cord is required to maximize its therapeutic efficacy in treating neuropathic pain.

Intranasal administration is emerging as a noninvasive drug delivery route that can transport drugs directly from the nasal cavity to the central nervous system, circumventing the blood–brain barrier [27]. We previously proposed a novel drug delivery system based on cell-penetrating peptide-modified polymer micelles created by chemically modifying a Tat peptide with a polyethylene glycol–polycaprolactone block copolymer (PEG-PCL-Tat) [28]. Subsequently, we demonstrated the efficacy of PEG-PCL-Tat as a nanocarrier for the intranasal administration of NAC in extending the lifespan in a murine model of amyotrophic lateral sclerosis (ALS), a neurodegenerative disorder characterized by the degeneration of spinal motor neurons [29]. Thus, the intranasal administration of therapeutic agents using PEG-PCL-Tat provides a noninvasive and effective alternative for localized delivery to the spinal cord, thereby providing a basis for its application to other neurological disorders. Building on our previous research in this study, we aimed to investigate whether the intranasal administration of NAC combined with PEG-PCL-Tat (NAC/PPT) could ameliorate neuropathic pain. Specifically, we examined whether post-onset NAC/PPT administration attenuates mechanical allodynia and abates pathophysiological alterations in glial cells within the spinal cord in a mouse model of partial sciatic nerve ligation (PSNL)-induced neuropathic pain.

## 2. Materials and Methods

### 2.1. Animals

Male ICR mice (Japan SLC Inc., Hamamatsu, Japan) weighing 25–30 g were housed at a temperature of 23 ± 1 °C with a 12 h light–dark cycle (light on 8:00 a.m. to 8:00 p.m.). Food and water were provided ad libitum. Mice were allowed to acclimate for 1 week before the study began. No signs of illness, abnormal behavior, or other health concerns were observed in any mice during the acclimatization period, and no animals were excluded from the study. The mice were then randomly assigned to the sham, PSNL, PSNL/NAC-PPT, and PSNL/NAC groups. Researchers were aware of the group assignments; however, all measurements and analyses were performed using standardized protocols to minimize bias. In total, 69 mice were allocated to the following three groups: 40/69 mice for the von Frey test, 20/69 mice for Western blotting, and 9/69 mice for immunohistochemistry (Figure 1). Notably, the von Frey test was performed to assess mechanical allodynia by measuring the mechanical withdrawal threshold of the hind paw (Section 2.5). Mice were anesthetized with isoflurane (4% induction and 2% maintenance) prior to tissue harvesting. The absence of a response to toe pinching indicated deep anesthesia. All animal experiments were conducted in accordance with the guidelines approved by the Nihon University Animal Care and Use Committee (Tokyo, Japan; experiment number #AP22PHA011-1).

### 2.2. Neuropathic Pain Model

PSNL mice were established as described by Seltzer et al. (1990) and Ito et al. (2022) [30,31]. Briefly, anesthesia was induced via the inhalation of 4% isoflurane in oxygen and maintained with 2% isoflurane in oxygen [32]. After anesthesia, the sciatic nerve trunk on the right side of the mice was exposed, and 1/3–1/2 of the sciatic nerve was ligated with a catgut 8-0 suture. Mechanical allodynia was assessed using the von Frey test after surgery, as detailed in Section 2.5. Sham surgeries were performed using the same procedure in the sham group, but nerve ligation was not performed. Finally, the mice were euthanized by cervical dislocation, and the spinal cords (L4–L6) were collected.

### 2.3. Preparation of the NAC Containing PEG-PCL-Tat Micelle Solution

PEG-PCL-Tat was synthesized as previously described [29,33,34] (Figure 2). Briefly, the Tat peptide and PEG-PCL block copolymer (each 0.02 mmol) were first dissolved in dimethylformamide. Equimolar amounts of water-soluble carbodiimide hydrochloride and 4-dimethylaminopyridine were then added, and the mixture was stirred for 24 h at 25 °C. The reaction mixture was dialyzed with a dialysis membrane (molecular weight cutoff: 3.5 kDa) against ultrapure water under gentle stirring for 24 h, followed by lyophilization to yield PEG-PCL–Tat. A 25 mg/mL PEG-PEL-Tat solution and a 100 mg/mL NAC solution were dissolved separately in 10 mM 2-[4-(2-hydroxyethyl)-1-piperazinyl] ethanesulfonic acid (HEPES) buffer (pH 7.4). Equal volumes of these solutions were mixed and incubated for 30 min at 25 °C to obtain 50 mg/mL NAC/PEG-PCL-Tat (NAC/PPT). Importantly, the prepared micelles demonstrated reproducible characteristics consistent with our previous report [31], exhibiting an average particle size of approximately 290 nm and a zeta potential of approximately +10 mV.

### 2.4. Mouse Intranasal Administration

Intranasal administration was initiated 8 days after surgery and was performed as previously described [29]. Briefly, mice were anesthetized with isoflurane (4% induction, 2% maintenance), and their nasal area was covered with an openable inhalation mask (SN-487-70-09, Shinano Seisakusho, Tokyo, Japan). After opening the silicone cap of the mask, a total volume of 20 µL of dosing solution was administered in 1 µL increments, alternating into each naris at 30 s intervals over 10 min (1 µL/30 s). The dosing solution was delivered via the placement of the microtip near the nasal cavity, with administration synchronized to the mouse’s breathing to allow for spontaneous aspiration. A total volume of 20 µL NAC/PEG-PCL-Tat solution per mouse was administered intranasally, equivalent to receiving 1.0 mg of NAC per mouse. NAC alone was administered at 1.0 mg per mouse. The sham group did not undergo any intranasal administration throughout the experimental period. Figure 3 illustrates the research protocol used in this study.

### 2.5. Assessment of Mechanical Withdrawal Thresholds

The withdrawal threshold (g) of the hind paw for mechanical stimulation was determined using von Frey filaments [35]. The sample size for the mice was determined by referencing previous studies [35]. Briefly, the von Frey filament was pressed against the mid-plantar surface of the hind paw, such that the filament was bent slightly. The lowest force that caused responses, such as the lifting and licking of the hind paw, was considered the withdrawal threshold. Both the left and right hind paws were tested three times, at 10 s intervals, and the mean withdrawal threshold was recorded. Withdrawal thresholds were measured at 3, 5, 7, 8, 9, 10, 11, 12, 13, and 14 days after PSNL or sham surgery (Figure 3).

### 2.6. Western Blot Analysis

Western blot analysis was performed as previously described [36]. Spinal cords (L4–L6) from five mice in each treatment group were used for Western blot analysis. The dissected spinal cords were lysed using a cell lysis buffer containing protease (Roche, Basel, Switzerland) and phosphatase (Sigma–Aldrich, St. Louis, MO, USA) inhibitors. Protein extracts were subjected to sodium dodecyl sulfate–polyacrylamide gel electrophoresis and transferred onto polyvinylidene difluoride membranes (Millipore, Burlington, MA, USA). The membranes were incubated with primary antibodies: anti-Iba-1 antibody (diluted 1:500; FUJIFILM Wako, Osaka, Japan) or anti-glial fibrillary acidic protein (GFAP) antibody (diluted 1:500; MERCK, Darmstadt, Germany) or anti-β-actin antibody (diluted 1:2000; Sigma–Aldrich) overnight at 4 °C. The membranes were washed repeatedly in Tris-buffered saline containing 0.05% *v*/*v* Tween 20 and incubated with a horseradish peroxidase-conjugated secondary antibody (Santa Cruz Biotechnology, Dallas, TX, USA, diluted 1:10,000) for 1 h. Immunoreactive bands were detected using an enhanced chemiluminescence detection system (Cytiva) with ImageQuant™ TL ver. 10.0 (Cytiva, Marlborough, MA, USA).

### 2.7. Immunohistochemistry

Briefly, spinal cord (L4–L6) tissue from at least three mice in each treatment group were cut into 30 µm-thick sections and incubated with rabbit anti-Iba-1 antibody (1:500; FUJIFILM Wako, Osaka, Japan) in Can Get Signal™ Immunostain Immunoreaction Enhancer Solution A (Toyobo, Osaka, Japan) at 4 °C for 72 h. After washing with phosphate-buffered saline, the sections were incubated with Alexa Fluor 488-conjugated goat anti-rabbit IgG secondary antibody (Molecular Probes, Eugene, OR, USA) (diluted 1:250) in the dark at room temperature for 2 h. Thereafter, the sections were mounted using VECTASHIELD^®^ HardSet™ Mounting Medium with DAPI (Vector Laboratories, Newark, CA, USA; concentration of DAPI, 1.5 µg/mL) for 24 h, and the nuclei were stained with DAPI. Fluorescence microscopy (BZ-X800, Keyence, Osaka, Japan) was performed to visualize and analyze the immunofluorescence staining of the Iba-1 positive microglia in the sections, as previously described [37].

### 2.8. Statistical Analysis

All statistical analyses were performed using GraphPad Prism 9 (GraphPad Software, La Jolla, CA, USA). Significant differences were analyzed using one-way or two-way analysis of variance (ANOVA), followed by Tukey’s test for multiple comparisons, as previously described [10]. Statistical significance was set at *p* < 0.05.

## 3. Results

### 3.1. Intranasal Administration of NAC/PPT Reduces the PSNL Surgery-Evoked Mechanical Allodynia

We first compared the analgesic effects of the intranasal administration of NAC and NAC/PPT in PSNL mice. As shown in Figure 3, treatment with NAC (1.0 mg) or NAC/PPT (1.0 mg) was initiated 8 days after PSNL, and mechanical allodynia was assessed using the hind paw withdrawal mechanical threshold in response to the von Frey test. A significant decrease in withdrawal thresholds in the hind paw ipsilateral to the surgery in the PNSL group compared to the Sham group was observed from days 3–14 post-surgery, with further exacerbation observed after day 11, but this was not seen in the contralateral hind paw (Figure 4). Treatment with NAC/PPT significantly ameliorated the PNSL-induced decrease in ipsilateral hind paw withdrawal thresholds on days 13 (95%CI: −0.2269 to −0.002851, *p* = 0.0429 vs. PNSL) and 14 (95%CI: −0.1279 to −0.01039, *p* = 0.0158 vs. PNSL) (Figure 4). In contrast, although NAC alone tended to improve the PNSL-induced exacerbation of mechanical allodynia from 11 to 14 days post-surgery, it did not exhibit any significant effect (Figure 4). Additionally, there was no significant difference between the NAC and NAC/PPT groups from 11 to 14 days post-surgery (11 days: 95%CI −0.03072 to 0.09205, *p* = 0.8221; 12 days: 95%CI −0.08977 to 0.2552, *p* = 0.7918; 13 days: 95%CI −0.01755 to 0.2064, *p* = 0.1334; 14 days: 95%CI −0.01652 to 0.1087, *p* = 0.3335); however, the NAC/PPT group tended to show more consistent improvement in withdrawal thresholds than the NAC group. These results demonstrated that utilizing PEG-PCL-Tat as a nanomicelle carrier enhanced the analgesic efficacy of NAC in the mouse PSNL-induced neuropathic pain model.

### 3.2. Intranasal Administration of NAC/PPT Demonstrated a Tendency to Alleviate Microglial Induction of the Spinal Cords in PSNL Mice

We performed Western blotting using antibodies against GFAP, an astrocyte marker, and Iba-1, a microglial marker, to evaluate the effects of the intranasal administration of NAC/PPT on the increase in astrocytes and microglia in the spinal cord of PSNL mice. As shown in Figure 5, GFAP expression did not change in any group (PSNL: 95%CI −13.47 to 46.67, *p* = 0.4173 vs. sham; 95%CI −53.26 to 6.878, *p* = 0.1636 vs. NAC/PPT; 95%CI −26.85 to 33.29, *p* = 0.9897 vs. NAC). In contrast, Iba-1 expression levels tended to increase in the spinal cords of PSNL mice (95%CI −158.8 to 73.31, *p* = 0.7214) compared to those in sham mice (Figure 6). Although treatment with NAC/PPT tended to suppress the PSNL-induced increase in Iba-1 expression to levels (95%CI −65.65 to 166.4, *p* = 0.6103) comparable to the sham group, NAC alone did not show this effect (Figure 6). Thus, using a homogenate of the whole spinal cord (L4–L6) containing the region responsible for neuropathic pain, Western blot analysis revealed the effect of NAC/PPT on microglial induction; however, no significant difference was observed.

### 3.3. Intranasal Administration of NAC/PPT Alleviated Microglial Induction of Spinal Dorsal Horn in PSNL Mice

We performed immunohistochemical analysis to determine the changes in the spinal dorsal horn microglia. Although the immunoreactivity of Iba-1 in the spinal dorsal horn ipsilateral to the surgery exhibited a significant increase in the PSNL group (95%CI −256.5 to −18.30, *p* = 0.0283) compared to that in the sham group, the intranasal administration of NAC/PPT significantly attenuated the PSNL-induced increase in Iba-1 expression on the ipsilateral side (NAC/PPT: 95%CI 2.592 to 240.8, *p* = 0.0461 vs. PSL, Figure 7). In the contralateral spinal dorsal horn, an increasing trend in Iba-1 expression was observed in PSNL mice, which was attenuated by NAC/PPT to levels comparable to those observed in the sham group (Figure 7). These findings suggest that NAC/PPT suppresses aberrant microglial induction in the dorsal horn of the spinal cord, particularly at the ipsilateral site of nerve injury in PSNL mice.

## 4. Discussion

We aimed to evaluate the potential of the post-onset intranasal administration of *N*-acetyl-L-cysteine (NAC) combined with PEG-PCL-Tat (NAC/PPT) for neuropathic pain and found that NAC/PPT significantly ameliorated mechanical allodynia in PSNL mice, even after disease onset. These findings underscore the efficacy of PEG-PCL-Tat as a delivery system to enhance the therapeutic potential of NAC in neuropathic pain.

Previous reports have shown that the intranasal administration of water-soluble substrates such as dextran and siRNA in combination with PEG-PCL-Tat result in a marked increase in brain distribution [28]. Moreover, studies have demonstrated the efficacy of PEG-PCL-Tat as a delivery system for siRNA to the brain via intranasal administration, with promising results in both malignant glioma [38] and cerebral ischemia-reperfusion injury rat models [34]. Based on these findings, our recent investigation [29] expanded the application of PEG-PCL-Tat to NAC, a small water-soluble molecule with potent antioxidant properties and low central nervous system permeability [22]. We found that the intranasal administration of NAC/PPT to ddY mice tended to increase the levels of NAC in the spinal cord and brain compared to the intranasal administration of NAC alone [29]. Notably, NAC/PPT also exhibited significant therapeutic effects in a mouse model of a spinal cord disease ALS, whereas neither NAC nor PEG-PCL-Tat alone showed any observable therapeutic benefits [29]. These results suggest that PEG-PCL-Tat may be useful as a drug delivery system to the brain and the spinal cord.

The activation of and increase in central glial cells such as astrocytes and microglia are widely acknowledged to play a pivotal role in the chronicity of pain by evoking neuroinflammation that sensitizes nociceptive sensory neurons [39]. Moreover, pain hypersensitivity induced by nerve injury is associated with abnormal excitability of spinal dorsal horn neurons, which occurs as a result of the activation and increase in microglia in the spinal dorsal horn [40]. Thus, owing to growing evidence of oxidative stress within the spinal cord in neuropathic pain, oxidative stress could be a potential pathogenic factor and therapeutic target for neuropathic pain [41]. Of particular note is the discovery that ROS enhance microglial activation in the spinal dorsal horn, whereas the inhibition of microglial ROS generation via intrathecal administration of the antioxidant sulforaphane attenuates mechanical allodynia and thermal hyperalgesia in nerve-injured mice [42]. The ratio of reduced glutathione/oxidized glutathione in the PSNL-induced neuropathic pain mouse model, which serves as a marker of oxidative stress and redox homeostasis, along with the level of malondialdehyde, a by-product of lipid peroxidation, was exacerbated in the spinal cord of PSNL mice at 7 and 14 days following PSNL surgery [43]. Furthermore, the levels of 4-hydroxy-2-nonenal-modified proteins and 8-hydroxy-2′-deoxyguanosine, which are a major end products of oxidative stress resulting from lipid peroxidation and DNA oxidation, respectively, have been reported to increase in the lumbar spinal cord dorsal horn [44]. Notably, the number of Iba-1-positive microglia was upregulated in the ipsilateral spinal dorsal horn of PSNL mice [45,46]. The chronic administration of minocycline, an inhibitor of microglial activation, effectively prevented the development of neuropathic pain and significantly reduced oxidative stress in rats models of CCI-induced neuropathy [47]. Collectively, these findings suggest that oxidative stress-associated microglial activation plays a pivotal role in mice with PSNL-induced neuropathy. Accordingly in this study, Western blot analysis of whole spinal cord (L4–L6) homogenates showed an increasing trend in Iba-1 expression: an increase in the PSNL group compared to the sham group and a decrease in NAC/PPT-treated PSNL mice compared to untreated PSNL mice. In contrast, NAC alone did not affect the trend of PSNL-induced increase in Iba-1 expression. Similarly, immunohistochemical analysis revealed that the intranasal administration of NAC/PPT significantly attenuated PSNL-induced increase in Iba-1 expression in the ipsilateral spinal dorsal horn, a region critical for sensory processing and pain pathology, following nerve injury [7]. These results suggest that NAC/PPT effectively suppresses oxidative stress-associated microglial expression in the spinal dorsal horn, leading to the attenuation of mechanical allodynia in PSNL-induced neuropathic pain models after onset. However, NAC has shown therapeutic effects beyond oxidative stress, effectively ameliorating CCI-induced neuropathic pain in rats by inhibiting matrix metalloproteinases-2 and -9 [26]. Further studies are needed to assess whether NAC/PPT exerts similar effects on microglial overexpression induced by non-oxidative stress factors.

An increase in reactive astrocytes typically follows microglial activation after nerve injury and remains in a reactive state during pathological pain [39]. However, Western blotting revealed that the expression of the astrocyte marker GFAP in the spinal cords remained unchanged among untreated PSNL mice and those treated with either NAC alone or NAC/PPT at 14 days post-surgery compared to that in the sham group. Similar to our findings, the Western blot analysis of the spinal cord in CCI rat models showed no increase in GFAP levels [26]. However, the oral administration of NAC from day 14 post-surgery in CCI rats demonstrated analgesic effects, accompanied by the suppression of increased protein expression of Iba-1 in the spinal cord [26]. Furthermore, continuous intrathecal infusion of NAC in rats with spinal cord injury suppressed the protein levels of the microglial marker cluster of differentiation 11b and 68 in the spinal cord, but not those of the astrocyte marker GFAP [48]. Taken together, these findings indicate that NAC exerts its analgesic effects in neuropathic pain by targeting microglia and not astrocytes, and suggest that astrocytes may play a less prominent role in sustaining the maintenance phase of pain in these neuropathic pain models. On the contrary, astrocytes reportedly play a protective role against neuropathic pain by decreasing synaptic glutamate [49]. Moreover, NAC exerts pro-survival effects on cultured astrocytes following oxidative stress and on pro-inflammatory cytokines [15,21,50]. Nonetheless, further studies are required to elucidate the potential action of NAC on both microglia and astrocytes and how this might contribute to its overall analgesic effect in neuropathic pain, with emphasis on whether NAC/PPT influences astrocyte polarization toward either cytotoxic or cytoprotective phenotypes.

Previous studies have demonstrated the analgesic effects of NAC in the CCI-induced neuropathic pain model rats and mice, with effective doses ranging from 100 mg/kg to 150 mg/kg administered either orally or intraperitoneally [24,25,26,51]. In this study, intranasal administration of NAC alone (1 mg/mouse, approximately 33–40 mg/kg based on–25–30 g mouse weight) had no analgesic effect, but the intranasal administration of NAC/PPT (1 mg/mouse) demonstrated significant analgesic effects. While the lack of a direct comparison within the same model requires cautious interpretation when discussing dosage effectiveness across different models, our results suggest that the intranasal administration of NAC/PPT provides analgesia at lower doses in PSNL mice than oral or intraperitoneal administration of NAC alone in previous CCI model studies [24,25,26,51]. Our previous study demonstrated that the intranasal administration of the NAC/PPT formulation not only enhanced NAC distribution in both the spinal cord and brain compared to NAC alone in ddY mice, but also exhibited significant therapeutic benefits in the ALS mouse model, while NAC alone showed no notable effects [29]. The enhanced efficacy of NAC/PPT on neuropathic pain and ALS compared with NAC alone suggests that PEG-PCL-Tat facilitates better absorption and transport of NAC from the nasal cavity to the central nervous system, potentially leading to increased bioavailability in the spinal cord. The intranasal administration of NAC or Alexa-dextran with PEG-PCL-Tat resulted in increased brain delivery via the olfactory and trigeminal nerve pathways [28,29]. Notably, the administered compounds initially travel along the perineural spaces surrounding the olfactory and trigeminal nerves and subsequently diffuse through perivascular spaces, enabling their extensive distribution to brain regions [52]. PEG modification enhances the colloidal stability of nanoparticles under diverse physiological conditions, such as variable ion concentrations and pH levels, thereby improving their diffusivity in the brain [53]. Additionally, neutral surface charge lipid nano-capsules exhibit advantages in mucus penetration, as the neutral charge minimizes electrostatic interactions with mucin, thus facilitating more effective drug delivery across mucus barriers compared to charged particles [54]. Similarly, our previous studies showed that PEG-modified liposomes with a near-neutral charge showed widespread distribution throughout the brain and spinal cord compared to charged liposomes, confirming these complementary features [55]. Notably, the NAC/PPT formulation used in this study, which consisted of equal volumes of 100 mg/mL NAC solution and 25 mg/mL PEG-PCL-Tat solution, exhibited a near-neutral charge [29]. Importantly, PEG-PCL-Tat contains a CPP, which enhances cellular membrane permeability [56]. Additionally, the arginine residues within the CPP sequence are critical for cell penetration [57]. Overall, these features of NAC/PPT likely contribute synergistically to mucus penetration and colloidal stability, facilitating efficient nose-to-brain/spinal cord delivery through the olfactory and trigeminal pathways.

To the best of our knowledge, none of the studies so far have examined the effects of intranasal formulations containing NAC on neuropathic pain. However, other intranasal NAC formulations, including those using silk fibroin nanoparticles or hyaluronic acid/dopamine/silk fibroin hydrogels, have demonstrated improved NAC delivery to brain tissues and the opening of tight junctions in nasal mucosal cells [58,59]. Compared to these systems, which largely focus on evaluating pharmacokinetics or delivery efficiency, our PEG-PCL-Tat micelles represent a novel therapeutic approach with robust in vivo validation in a neuropathic pain model. Nevertheless, it is crucial to acknowledge that a direct comparison with other intranasal formulations has not yet been conducted. Further studies under identical experimental conditions are warranted to provide a more comprehensive understanding. Furthermore, although the initial nose-to-brain delivery mechanism is well understood, the subsequent distribution mechanism from brain to spinal cord warrants further investigation. Texas Red-conjugated dextran administered into the cisterna magna demonstrated spinal cord distribution through the brain surface, perivascular spaces, and glymphatic clearance pathway [60]; however, whether PEG-PCL-Tat follows similar or distinct routes remains unclear. Therefore, further studies comparing its efficacy, pharmacokinetics, and bioavailability, both with orally or intraperitoneally administered NAC alone and with other intranasal NAC formulations in the same model, are warranted to fully establish the therapeutic benefits of intranasal NAC/PPT in neuropathic pain treatment, particularly to clarify the potential advantages offered by the brain-to-spinal cord distribution pathway.

## 5. Conclusions

In this study, we demonstrated that the intranasal administration of NAC/PPT, but not NAC alone, significantly ameliorated mechanical allodynia in a mouse model of PSNL-induced neuropathic pain by suppressing microglial induction in the spinal dorsal horn. Notably, NAC/PPT showed efficacy at lower doses than those in previous studies using oral or intraperitoneal administration of NAC, suggesting improved bioavailability and targeted delivery to the spinal cord. Our findings highlight the potential of intranasal drug delivery systems using PEG-PCL-Tat as a novel therapeutic approach for the treatment of neuropathic pain. The enhanced efficacy of therapeutics using this system opens new avenues for targeted drug delivery to the central nervous system, potentially improving treatment outcomes in neuropathic pain conditions.

## Figures and Tables

**Figure 1 pharmaceutics-17-00044-f001:**
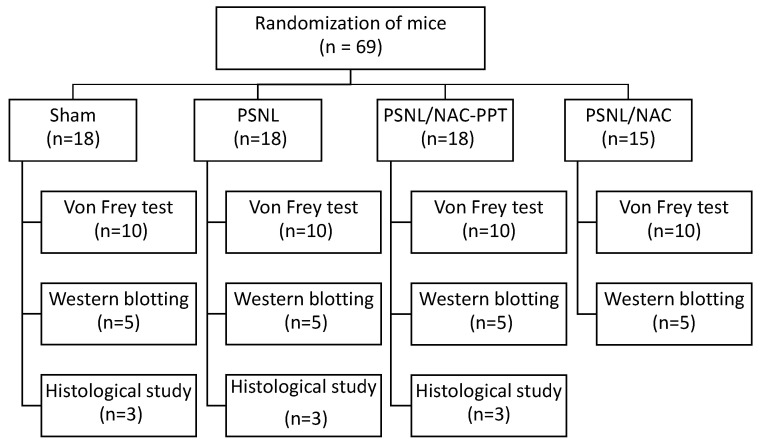
Flowchart of the study design. PSNL: partial sciatic nerve ligation, NAC: *N*-acetyl-L-cysteine, NAC-PPT: NAC combined with PEG-PCL-Tat.

**Figure 2 pharmaceutics-17-00044-f002:**
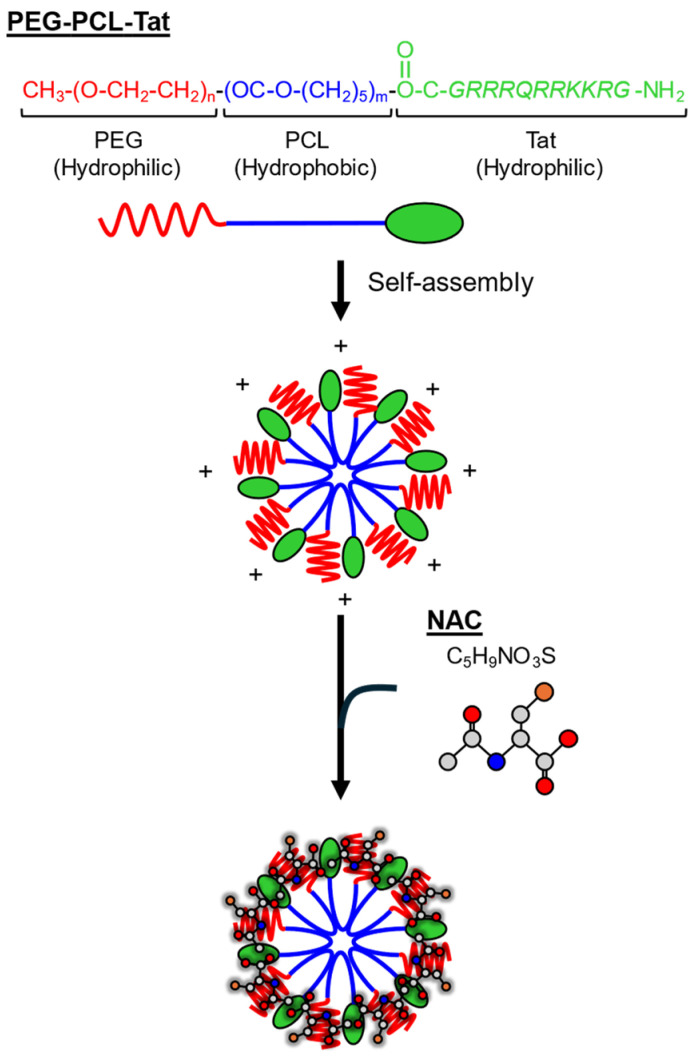
Schematic illustration of *N*-acetyl-L-cysteine/PEG-PCL-Tat (NAC/PPT) micelle formation. The amphiphilic block copolymer PEG-PCL-Tat consists of three distinct segments: a hydrophilic PEG shell (shown in red wavy lines), a hydrophobic PCL core (shown in blue straight lines), and a cell-penetrating Tat peptide (shown as green ovals). The polymer undergoes self-assembly in an aqueous solution to form micelles with a positively charged surface. NAC (represented by atoms in grey dots for carbon, red dots for oxygen, blue dot for nitrogen, and orange dot for sulfur) is incorporated into the micelle structure through electrostatic interactions, forming NAC/PPT micelles.

**Figure 3 pharmaceutics-17-00044-f003:**
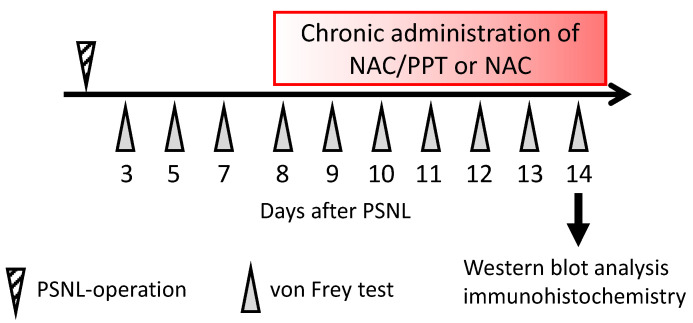
Research protocol followed in this study. Eight days following PSNL surgery, NAC/PPT or NAC was administered intranasally. Hind paw withdrawal thresholds were measured on the indicated days after PSNL or sham surgery. Fourteen days after the PSNL surgery, whole spinal cords (L4–L6) were subjected to Western blot and immunohistochemistry.

**Figure 4 pharmaceutics-17-00044-f004:**
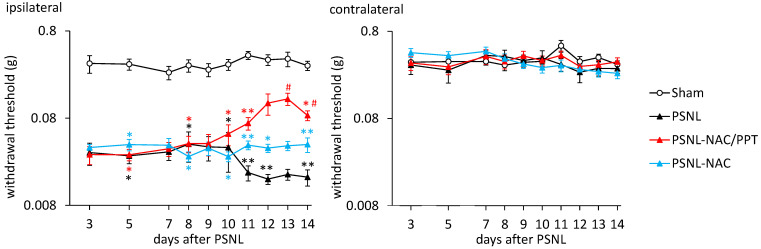
Effect of *N*-acetyl-L-cysteine combined with PEG-PCL-Tat (NAC/PPT) on partial sciatic nerve ligation (PSNL)-induced mechanical allodynia. Mice were intranasally administered NAC (1.0 mg) or NAC/PPT (1.0 mg), starting at 8 days post-PSNL surgery. Graphs show mechanical thresholds of hind paw withdrawal responses ipsilateral (**left**) and contralateral (**right**) to PSNL surgery in reaction to von Frey filaments, monitored 3–14 days after PSNL. Data are shown as mean ± standard error, calculated from ten independent biological replicates. Statistical significance was determined using two-way ANOVA, followed by the Tukey’s test (ipsilateral; interaction, F_27, 324_ = 1.148, *p* = 0.2827; time F_4.162, 149.8_ = 1.598, *p* = 0.1755, and treatment, F_3, 36_ = 116.1, *p* < 0.0001; contralateral; interaction, F_27, 324_ = 1.137, *p* = 0.2943; time, F_6.400, 230.4_ = 1.979, *p* = 0.0650, and treatment, F_3, 36_ = 0.3475, *p* = 0.7911). * *p* < 0.05, ** *p* < 0.05 compared to the Sham group; ^#^ *p* < 0.05 compared to the corresponding PSNL group.

**Figure 5 pharmaceutics-17-00044-f005:**
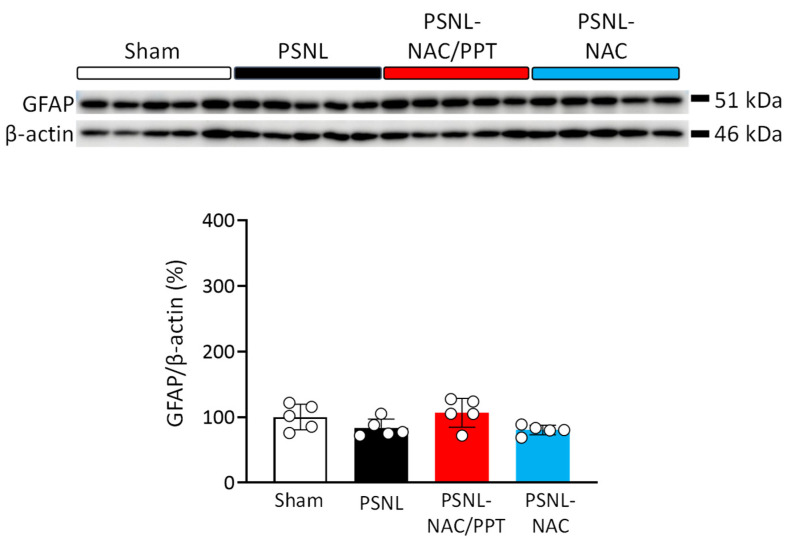
Effect of intranasal treatment with *N*-acetyl-L-cysteine (NAC) combined with PEG-PCL-Tat (NAC/PPT) on partial sciatic nerve ligation (PSNL)-induced astrocyte expression in spinal cords. Mice were intranasally administered NAC (1.0 mg) or NAC/PPT (1.0 mg), starting at 8 days post-PSNL surgery. Fourteen days after the PSNL surgery, whole spinal cords (L4–L6) were subjected to Western blot analysis. Photographs show Western blot signals for glial fibrillary acidic protein (GFAP) with β-actin as an internal control. Graphs show the ratio of band intensities for GFAP, normalized to β-actin levels, and expressed relative to the sham group. Data are shown as mean ± standard deviation, calculated from five independent biological replicates. Significant difference was determined by the one-way ANOVA followed by the Tukey’s test. No significant differences were detected among the groups (*p* > 0.05).

**Figure 6 pharmaceutics-17-00044-f006:**
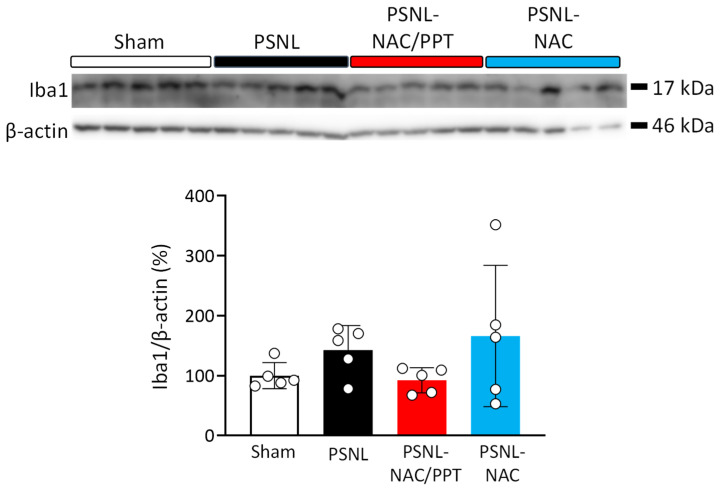
Effect of intranasal treatment with *N*-acetyl-L-cysteine (NAC) combined with PEG-PCL-Tat (NAC/PPT) on partial sciatic nerve ligation (PSNL)-induced microglial expression in spinal cords. Mice were intranasally administered NAC (1.0 mg) or NAC/PPT (1.0 mg), starting at 8 days post-PSNL surgery. Fourteen days after PSNL surgery, whole spinal cords (L4–L6) were subjected to Western blot analysis. Photographs show Western blot signals for ionized calcium-binding adapter molecule 1 (Iba-1), microglial marker, with β-actin as internal control. Graphs show ratio of band intensities for Iba-1, normalized to β-actin levels, and expressed relative to sham group. Data are shown as mean ± standard deviation, calculated from five independent biological replicates. Significant difference was determined by one-way ANOVA followed by Tukey’s test. No significant differences were detected among groups (*p* > 0.05).

**Figure 7 pharmaceutics-17-00044-f007:**
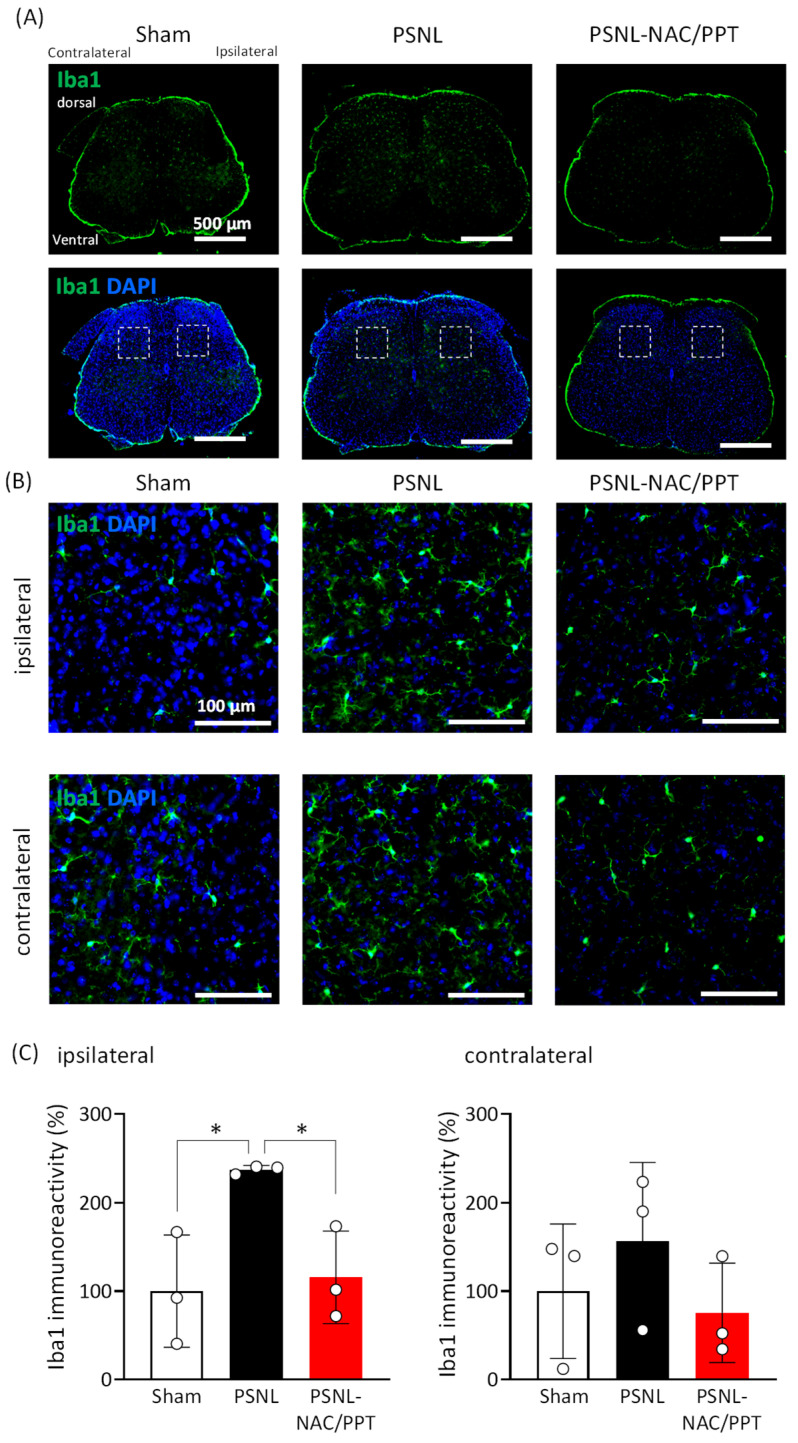
Effect of intranasal treatment with *N*-acetyl-L-cysteine combined with PEG-PCL-Tat (NAC/PPT) on partial sciatic nerve ligation (PSNL)-induced increases in microglial population in spinal dorsal horn. Mice were intranasally administered NAC/PPT (1.0 mg), starting at 8 days post-PSNL surgery. Fourteen days after PSNL surgery, spinal cords (L4–L6) were subjected to immunohistochemistry. (**A**) Photographs show representative confocal images of immunofluorescence staining for Iba-1 in the entire spinal cord (L4–L6). The white dashed box area was captured at high magnification to obtain photographs (**B**). Scale bar is 500 μm. (**B**) Photographs show representative confocal images of immunofluorescence staining for Iba-1 in dorsal spinal cord (L4–L6), in manner ipsilateral (the upper panel) and contralateral (the **lower** panel) to the PSNL surgery. Scale bar is 100 μm. (**C**) Graphs show semiquantitative analysis of changes in Iba-1 immunoreactivity in ipsilateral (the **left** panel) and contralateral (the **right** panel) dorsal spinal cord (L4–L6), expressed relative to sham group. Data are shown as mean ± standard deviation, calculated from three independent biological replicates. Significant difference was determined by one-way ANOVA, followed by Tukey’s test. * *p* < 0.05.

## Data Availability

The original contributions presented in this study are included in the article. Further inquiries can be directed to the corresponding authors.

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
