# Peer review of "Therapeutic Efficacy of Intranasal *N*-Acetyl-L-Cysteine with Cell-Penetrating Peptide-Modified Polymer Micelles on Neuropathic Pain in Partial Sciatic Nerve Ligation Mice"

_pharmaceutics, 2025, doi:10.3390/pharmaceutics17010044_

Round 1
Reviewer 1 Report
Comments and Suggestions for Authors
Nango et al. studied the efficacy of intranasal delivery of N-acetyl-L-cysteine (NAC) in combination with PEG-PCL-Tat (NAC/PPT) for the treatment of neuropathic pain in mice. The findings indicated that NAC/PPT markedly decreased pain and microglial activation in the spinal cord, but NAC alone exhibited no significant impact, indicating that NAC/PPT may serve as a potential therapeutic option for neuropathic pain. The study is interesting; it was properly designed. The employed techniques are feasible, and the analysis and interpretation of the results facilitate a conclusion that is clearly substantiated by the findings. The authors adhere to the limit on self-citations and include current bibliographic references that corroborate and justify the conclusions of this study. This original study article may interest the scientific community as it presents pertinent information on the favorable effects of (NAC/PPT) and its potential clinical application in patients with neuropathy. Nonetheless, as indicated by the authors, additional study is required to validate the prospective function of the NAC/PPT and its influence on neuropathic pain. To enhance the manuscript's quality, it is essential to substantially refine its presentation, particularly with the description and analysis of the results and the accompanying figures. References must adhere to a consistent format, and it is advisable to incorporate more recent sources. I have following minor suggestions:
1. In figure 4, i suggest the authors to include baseline for all the groups before sciatic nerve surgery, and indicate surgery day and then behavior test results post surgery for easier interpretation of results.
2. For Fig.7, it is essntial to include whole section of the spinal cord and then mark the ipisilateral and contralateral areas to check the glial cell activation.
Reviewer 2 Report
Comments and Suggestions for Authors
pharmaceutics-3381346
Therapeutic Efficacy of Intranasal N-Acetyl-L-cysteine with Cell-Penetrating Peptide-Modified Polymer Micelles on Neuropathic Pain in Partial Sciatic Nerve Ligation Mice
The manuscript by Nango et al. described the in vivo evaluation of previously developed polymeric micelles for delivering NAC to the CNS via the IN route. The manuscript was well organized and the data were partly sufficient for the publication. Below are some minor suggestions to improve the manuscript.
1. Introduction: The first 3 paragraphs can be shortened. Since this work is based on a previously developed formulation by the same author group, the authors should expand paragraph 4 by briefly summarizing the previous results and explaining how this work can extend the previous work.
2. Figures: Please revise the graphs in Figure 4 by making the X-axis following the time scale.
3. Discussion: Please compare the results with those of other IN NAC-loaded formulations to highlight the strengths of the current formulation. Please discuss the limitations of this work/ formulation.
Round 2
Reviewer 1 Report
Comments and Suggestions for Authors
The authors have addressed majority of my concerns. I do not have any further comments.
Reviewer 2 Report
Comments and Suggestions for Authors
The manuscript was appropriately revised and can be accepted as is.